# Molecular conformation of polyelectrolytes inside Layer-by-Layer assembled films

Philipp Gutfreund [1] ✉, Christophe Higy[1,2], Giovanna Fragneto [1,3], Michel Tschopp[2], Olivier Felix [2] & Gero Decher [2]

Among all methods available for the preparation of multifunctional nanostructured composite materials with remarkable functional properties, Layer-by-Layer (LbL) assembly is currently one of the most widely used techniques due to its environmental friendliness, its ease of use and its versatility in combining a plethora of available colloids and macromolecules into finely tuned multicomponent architectures with nanometer scale control. Despite the importance of these systems in emerging technologies, their nanoscopic 3D structure, and thus the ability to predict and understand the device performance, is still largely unknown. In this article, we use neutron scattering to determine the average conformation of individual deuterated polyelectrolyte chains inside LbL assembled films. In particular, we determine that in LbL-films composed of poly(sodium 4-styrenesulfonate) (PSS) and poly(allylamine hydrochloride) (PAH) multilayers prepared from 2 M sodium chloride solutions the PSS chains exhibit a flattened coil conformation with an asymmetry factor of around seven. Albeit this highly non-equilibrium state of the polymer chain, its density profiles follow Gaussian distributions occupying roughly the same volume as in the bulk complex.

Layer-by-Layer (LbL) assembly was introduced in the early 1990's[1,2] as a scalable method for constructing multi-composite nanometric films without highly specialized or expensive equipment. Based on attractive interactions between a surface and molecules or colloids in solution the film forming components are deposited by classic adsorption to form fairly stratified, albeit fuzzy multilayer structures. The use of coulombic interactions for each adsorption step became highly popular due to the large number of available anionic or cationic components and the ease by which the electrostatic interactions can be fine-tuned for example by adjusting the pH or by adding salt. In addition to bringing a solution into contact with a surface by immersion/withdrawal cycles (dipping), fabrication friendly methods like spray-assisted LbL assembly or spin-assisted LbL assembly can be used as well. In the interest of conserving space we reference a few reviews that provide extended details about the components that can be used in LbL-assembly, the interactions between them, the resulting structures and morphologies and their tunability and the potential applications[2–18].

Polyelectrolyte multilayers (PEMs) are non-equilibrium structures composed of polyanions and polycations which, depending on their mobility in a specific polyelectrolyte complex (PEC), are either trapped in the Z-position at which they are deposited (solid-like PEC)[19] or which can diffuse throughout the film (coacervate-like or liquid-like PEC) bringing the PEM closer to equilibrium. The mobility of a polyelectrolyte (PE) in a PEC depends predominately on the chemical nature of the PE, its molar mass, its charge density along the chain, the chemical nature of added salt, the ionic strength and the water content. In this work we will deal with the molecular structure of poly(sodium 4-styrenesulfonate) (PSS) in multilayers with poly(allylamine hydrochloride) (PAH), a very common system, believed to be a solid complex.

While some work has been done to investigate the chain conformation in bulk PECs[20,21] and some coil dimensions have been reported, it is not at all clear what the PE chain conformation looks like in a solid-like PEM in the initial state after deposition and without any post-preparation treatments that would bring it closer to

[1]Institut Laue-Langevin, 71 avenue des Martyrs, 38042 Grenoble, France. [2]Institut Charles Sadron, Université de Strasbourg, 67034 Strasbourg, France. [3]Present address: European Spallation Source ERIC, P.O. Box 176, 22100 Lund, Sweden. ✉e-mail: gutfreund@ill.eu

equilibrium[22]. What is known, is that neighboring layers are highly overlapping, which was largely determined by extracting the scattering length density (SLD) profiles from neutron and X-ray reflectometry (NR and XRR, respectively). In a first approximation one assumes a 1:1 charge stoichiometry in the multilayer which would correspond to a homogeneous distribution of polyanions and polycations within the film. Early NR data have supported such a model, Bragg peaks indicating the presence of stratified layers of deuterated PSS (d-PSS) were only observed if the spacing between two d-PSS layers was at least 3 layers[2]. In the present work high-flux NR was used[23,24] and it is shown that the distribution of the deuterated polyanion is not homogeneous along the layer normal. This necessarily implies that locally the 1:1 charge stochiometry is not obeyed, but oscillates within one percent, testifying the non-equilibrium nature of the complex.

Concerning the PSS chain conformation, it is known (for example from ellipsometry or XRR) that the thickness increment per layer is much less than the unperturbed size of the PE chain and depends on the salt concentration of the used solution as well as on the deposition method. This testifies that the shape of a polyelectrolyte after adsorption is necessarily different from the bulk PEC case and that the adsorbed polymer chain in solid-like PECs remembers, to some extent, from which solution it was deposited. Neutron scattering is currently the only method capable of extracting the structure of single polymer chains in a crowded environment of identical molecules with sub-nanometer resolution. This technique was used here to reveal the molecular conformation of d-PSS in (PSS/PAH)$_n$ multilayers using a large number of film architectures. Understanding the general nanoscopic structure of new materials is a key point in predicting device performance (functionality) as well as trust and value with respect to applications and societal impact.

## Results

### Conformation of PSS chains normal to the surface (specular neutron reflectometry)

In order to investigate the conformation of PSS chains in direction perpendicular to the surface we performed an exhaustive specular NR study on samples prepared by dipping, spray- or spin-assisted assembly. For each preparation technique the total number of deposited layer pairs as well as the number and position of deuteration-labeled PSS (d-PSS) layers were varied in order to provide enough scattering contrasts for an unambiguous interpretation of the data. In total more than 50 samples were investigated on five different reflectometers on three different neutron sources to minimize systematic errors and get statistically relevant data. Only few scattering curves are presented in the main article, the vast majority can be found in the Supplementary Information.

The minimum total number of layer pairs used was ten with the majority of samples being over 24 layer pairs in order to mitigate the effect of surface layers, namely the fact that the first and last few layer pairs show different thickness than the bulk layers in the middle of the stack[25]. In the following only structural properties originating from these bulk zones will be discussed. Moreover, all investigated samples have an even number of PAH and PSS layers with always a PAH layer on top in order to avoid any influence of the so-called odd-even effect[11].

Additionally to the large amount of samples investigated we employed a global data fitting approach using a minimum amount of free fitting parameters[26]. This can be appreciated in the Supplementary Figures 2 and 3 where the reflectivities of 16 samples prepared in the same conditions, by spraying of solutions containing 0.5 M NaCl but with different positions of deuterium-labeled layers, are fitted using one global parameter for the effective SLD of the deuterated strata (SLD$_d$) and protonated layers (SLD$_h$) and a single parameter for all interlayer roughnesses $\sigma$. Only the layer pair thickness $d$ had to be adjusted for each sample individually in order to obtain a reasonable fit, which is due to the very high resolution of NR on layer pair

**Table 1 | Fitted layer pair thickness $d$, interlayer roughness $\sigma$, out-of-plane radius of gyration of the PSS ($R_g^{\perp}$), scattering length density of the deuterated PSS layer (SLD$_d$) and the protonated strata (SLD$_h$) deduced from NR**

| Deposition method | NaCl [M] | $d$ [Å] | $\sigma$ [Å] | $R_g^{\perp}$ [Å] | SLD$_d$ [$10^{-6}$Å$^{-2}$] | SLD$_h$ [$10^{-6}$Å$^{-2}$] |
|---|---|---|---|---|---|---|
| Spray-assisted | 0.5 | 26 ± 1 | 14 ± 2 | 15 ± 2 | 2.7 ± 0.3 | 1.2 ± 0.1 |
| Spin-assisted | 2 | 34 ± 2 | 15 ± 2 | 16.5 ± 2 | 2.9 ± 0.2 | 1.5 ± 0.2 |
| Spray-assisted | 2 | 41 ± 2 | 20 ± 3 | 21.6 ± 4 | 2.7 ± 0.3 | 1.1 ± 0.3 |
| Dipping | 2 | 52 ± 1 | 23 ± 3 | 25.3 ± 3 | 3.1 ± 0.1 | 1.1 ± 0.3 |

thickness of better than 0.5 Å revealing very subtle differences between samples, probably linked to their preparation and slightly different ambient humidity during the NR measurements. The resulting parameters using the average layer pair thickness are summarized in Table 1 (together with the results from the other preparation conditions).

Overall, the layer pair thickness is well in agreement with previous studies on dipped[26] and sprayed[27] PSS/PAH multilayers. A clear dependence on the preparation method can be observed at constant ionic strength. The thickness relates as: Spin-assisted assembly < Spray-assisted assembly < Dipping. The fact that sprayed films exhibit thinner layer pairs (and smaller interlayer roughness) than dipped ones was already observed before by NR[27]. In general, films prepared by spin-assisted LbL-assembly were also shown to be thinner (and with smoother internal interfaces) than the ones prepared conventionally in immersion/withdrawal cycles[28]. Increased layer pair thickness with ionic strength was also reported earlier for dipped films[26] with the thickness increasing by about 16 Å per 1M NaCl. In our study we measured only at two salt concentrations (0.5 M and 2 M), but if we assume a linear relationship as well we get a proportionality factor of about 10 Å per 1M NaCl for sprayed films.

Concerning interlayer roughness, the fact that the roughness between subsequent layers is constant for all interfaces and amounts to about half of the layer pair thickness was already revealed before by NR[26], where by varying the ionic strength of the dipping solution a linear relationship of 0.4 between layer pair thickness and roughness of PSS/PAH multilayers was revealed. When looking at the resulting roughness reported for all sample series in Table 1 it can be seen that this relationship between roughness and layer thickness holds even for the different preparation conditions showing an even more general meaning of this observation. When looking at the SLD profile normal to the surface of a single PSS layer (see Fig. 1b) it becomes evident that the resulting SLD profile corresponds to a Gaussian distribution, which is the expected profile assuming a random walk of a fully flexible polymer chain. This explains why the roughness and the thickness of the PSS layers are not independent variables but set by the width of the Gaussian function, which in turn is related to the radius of gyration of the polymer chain as explained in the Methods section Equation (1). This testifies two important results: (a) The layers correspond to molecular monolayers as was already argued in many reports on PEMs and (b) The polymer segments exhibit a Gaussian density distribution normal to the surface despite the fact that the chain is highly deformed for all preparation conditions tested here. Figure 1a also shows for the first time a clear Bragg peak for a PEM where every polyanion layer is labeled, clearly testifying that the PSS monomer density profile is not completely flat but oscillates as expected as can be seen in the SLD profile in Fig. 1b. The reason why this was not observed before is likely due to the sensitivity of the NR technique, which has increased nowadays due to recent instrumental developments[23,24] allowing for the detection of a Bragg peak with a maximum peak intensity below $10^{-6}$ as seen in Fig. 1a. Also well-controlled, reliable and automated preparation allowed to prepare samples with more than 70 layer pairs enhancing the Bragg-peak intensity. We note here that the fact that the

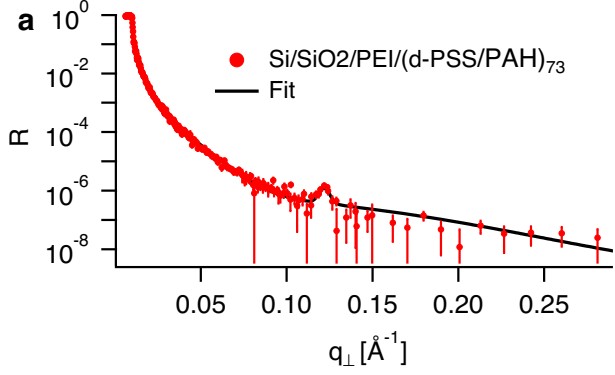

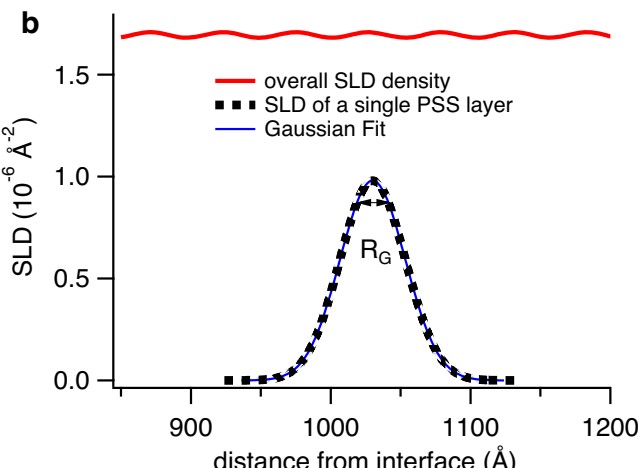

**Fig. 1 | Specular neutron reflectometry. a** NR (log scale) of a d-PSS/PAH multilayer with 73 layer pairs prepared by dipping from 2 M NaCl solutions (red points, error bars showing the statistical counting error). The black solid line corresponds to a fit to the data. Data were collected on the D17 reflectometer at the ILL. **b** Part of the SLD profile (red solid line) corresponding to the fit to the NR curve from (**a**) (Fitting parameters can be found in the Supplementary Table. 2). The broken black line is the SLD profile of a single PSS layer and the blue solid line is a Gaussian fit to the former. The small arrow underneath the maximum of the broken line indicates the radius of gyration of the Gaussian fit.

PSS monomer density is not completely flat means that the 1:1 charge stochiometry is locally not obeyed. From the SLD profile one can conclude that the peak-to-peak deviation of the PSS density from average is around 1% in the here studied case.

Looking now at the nominal SLD values reported in Table 1 we can observe that the SLDs for the deuterated and protonated strata are in accord with previous values reported for dipped[26] and sprayed[27] PSS/PAH samples in air. However, we present for the first time a direct comparison of three preparation conditions allowing for an additional trend to be revealed: the fitted SLD values of the deuterated and protonated strata are closer to each other for the sprayed and spin-coated samples compared to the dipped films. This cannot be explained by a difference in hydration. The only plausible explanation is that some defects must be present in the stratification allowing for the intermixing of deuterated and protonated polymers beyond the Gaussian-type roughness at the polymer/polymer interface, at least for sprayed and spin-coated samples. This is likely due to some kind of agglomeration, which will be discussed later in relation to the in-plane structure.

## Conformation of a polyelectrolyte chain in the direction parallel to the surface

In this section we compare the results obtained by Grazing Incidence Samll Angle Neutron Scattering (GI-SANS) and transmission

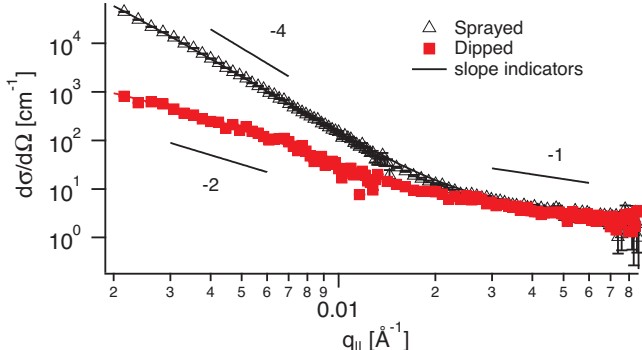

**Fig. 2 | Small Angle Neutron Scattering.** Transmission SANS cross-sections for a dipped film (red squares, 75 layer pairs) and a sprayed film (black triangles, 80 layer pairs) on a log-log scale, both prepared in 2 M NaCl solutions. The lines going through the data points are fits as explained in the main text; the other lines are slope indicators. Error bars show the statistical counting error.

SANS measurements on LbL films. We will use the unique possibility of SANS to access the form of single PSS chains in the highly crowded state of the thin solid film by labeling some of the PSS chains by deuteration[29]. This is routinely done for bulk samples and we shall systematically compare the in-plane structure of the here studied thin film to the bulk scattering curves of similar PSS complexes[21,30,31]. For these investigations we will use films produced at relatively high salt concentration (2 M NaCl) in order to increase the film thickness as the scattering intensity scales with the square of the film thickness and is generally very low in these type of measurements. We note that even at this salt concentration electrostatic interactions are not completely screened as the radius of gyration of a free PSS chain at this concentration is still roughly twice as large as in theta conditions[20,21].

Looking at the SANS curves (Fig. 2) of the dipped and sprayed samples a clear trend can be observed: At high q-values all curves show a common power law with an exponent between -1 and -1.5 and the same SANS absolute cross-section. This is commonly observed for polymer melts and is a measure of the chain stiffness. As all SANS-samples were produced at the same salt concentration it is not surprising that the short-range interactions are screened the same for all samples.

At low q, however, clear differences can be observed between the samples. While the dipped samples show power laws with exponents between -2 and -2.8, with absolute intensities on the order of 10 · 800 cm⁻¹, the sprayed sample cross-sections turn directly into a much steeper power law behavior between -2.5 and -3.8, indicating larger fractal structures, with absolute SANS intensities going from 50 cm⁻¹ up to 70000 cm⁻¹.

Given the strong fractal scattering of the sprayed samples it is difficult to extract the size of the single chains from this data in a model independent way as the single chain form factor scattering is buried below the signal from the much larger fractal structures. For the dipped samples, however, we can attempt to extract the radius of gyration in in-plane direction $R_g^{\parallel}$ by performing a Guinier analysis. In Fig. 3 the logarithm of the SANS intensity is plotted versus the square of the momentum transfer in the q-range $>1/R_g^{\parallel}$ and $<3/R_g^{\parallel}$. The slope yields an in-plane radius of gyration of $R_g^{\parallel} = 180 \pm 10$ Å for a PSS chain with a molecular weight of 80,800 g/mol.

As mentioned above at larger q-values the scattering function of a single polymer chain changes its slope as gradually internal structures of the polymer chains become visible. The change-over between the slopes of around $q^{-2}$ from a flexible chain to $q^{-1}$ for a stiff chain is often used to extract the persistence length[29,32], a measure to which extent the chain still feels its finite stiffness. When looking at the curves in

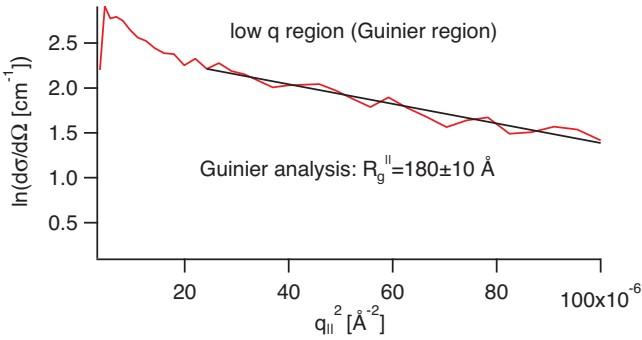

**Fig. 3 | Determination of the single chain radius of gyration.** Guinier plot of the dipped sample scattering cross-section from Fig. 2. The slope is fitted by the solid line indicating a radius of gyration of $180 \pm 10$ Å.

Fig. 2 this change-over happens around 0.024 Å$^{-1}$, at least for the dipped sample where the single chain scattering is not overwhelmed by the scattering of fractals as for the other curve. This value is significantly lower than for free PSS chains in solution, where this change-over happens around 0.05 Å$^{-1}$ for salt-free solutions[29] or at even larger values for salted PSS solutions[32], pointing towards a stiffer in-plane chain state in case of LbL films.

Quantitative fits to the SANS curves (lines in Fig. 2) corroborated the results extracted from the model free analysis, please refer to the Supplementary Information for more details. We also performed GISANS measurements on dipped samples (see Supplementary Information), they revealed basically the same results as transmission SANS with low q fractal scattering and a radius of gyration between 170 Å and 180 Å for the higher q scattering. The difference to transmission SANS is that the PEM layers closer to the surface contribute more to the scattering as compared to the buried ones and therefore a difference in structure between layers close to the surface as compared to bulk ones could have been observed, which was apparently not the case here.

**Roughness correlations at the polyelectrolyte interfaces as revealed by OSS**

In this section we analyze the results of off-specular neutron scattering (OSS) for an LbL film prepared by dipping. In Fig. 4a the OSS map for this sample is shown and a clear horizontal streak at $q_\perp = 0.07$ Å$^{-1}$ is visible, indicating a Bragg sheet. Such a feature, often visible in lamellar multilayers, points towards the existence of correlated in-plane roughness at the d-PSS/PAH interfaces. If the PEs were interdigitated on a monomer level, as is the case for miscible neutral polymers for example, the absence of any in-plane correlation of the interfacial structure would suppress all OSS scattering outside the mirror reflection for the momentum transfer space covered by the detector. For PEMs, on the contrary, we observed this correlated roughness for all investigated samples, including sprayed ones.

A quantitative analysis of the OSS pattern within the Distorted Wave Born Approximation (DWBA)[33] indeed revealed that the in-plane roughness between PSS and PAH layers was correlated and if a Gaussian correlation function is assumed then a radius of gyration of $180 \pm 100$ Å could reproduce the shape of the Bragg sheet as seen in Fig. 4b. This means that the PEMs are not interdigitated on a monomer level as is the case for miscible polymers, but on a molecular level hinting towards some kind of molecular phase separation at the polymer/polymer interface. Moreover, in order to reproduce the absolute intensity of the Bragg sheet, the roughness had to be modeled also correlated out-of-plane with a correlation length on the order of the total thickness of the sample meaning perfect out-of-plane correlation, similar to the electrostatic correlation peak observed in small angle scattering on PE solutions or complexes.

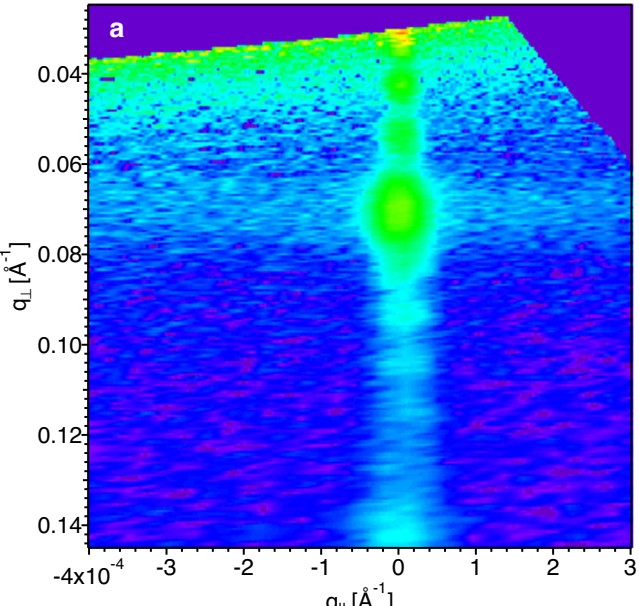

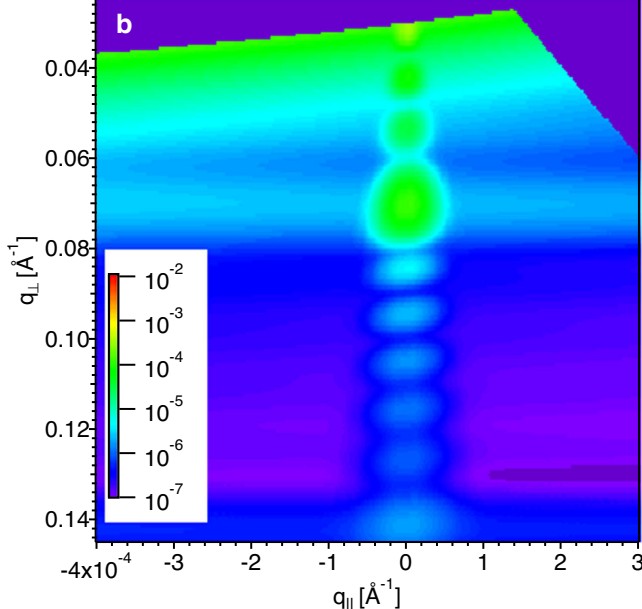

**Fig. 4 | Polyanion/polycation interface correlations. a** OSS map of a dipped sample from 2 M NaCl solutions (reflected intensity on logarithmic color scale). The sample consists of 27 layer pairs with every 4th PSS layer deuterated. **b** DWBA simulation assuming in-plane structural correlation as described in the main text. Both maps are drawn with the same color scale and identical axes.

## Discussion

In case of a sample prepared by dipping from a 2 M NaCl solution the radius of gyration of a PSS chain in the direction parallel to the surface is equal to $180 \pm 40$ Å whereas the radius of gyration in the direction perpendicular to the surface is equal to $25 \pm 3$ Å, which is 7 times smaller. This shows for the first time directly that the PSS chains with a molecular weight of $M_w = 80,800$ g/mol in the dipped film prepared in the conditions described above have a flattened coil conformation.

In comparison, from SANS studies on bulk PSS complexes with poly(diallyldimethylammonium chloride) (PDADMA) we can estimate the bulk radius of gyration of the PSS in a complex to be around 90 Å in our case (see detailed estimation of the bulk $R_g$ in the Supplementary Information). Now we can compare the volume occupied by one chain

in the two complexes by taking the bulk $R_g$ as the radius of a sphere and $R_g^\perp$ and $R_g^\parallel$ for the axes of an ellipsoid in the LbL film. This gives a PSS gyration volume of $3*10^6$ Å$^3$ for the bulk chain in a PDADMA complex and $3.3*10^6$ Å$^3$ in the here studied multilayer, which are identical within the experimental uncertainties.

Interestingly the deformation of the PSS molecule is not limited to large distances on the order of the entire molecule, reflected in the radius of gyration, as is the case for solvent-mediated swelling or shear-induced deformation[34], but is also visible in the effective monomer size and the persistence length as well. Similar to the twofold increase of the radius of gyration from 90 Å in the bulk to 180 Å in-plane radius in the here studied dipped LbL films, the persistence length also increased from around 50 Å to 90 Å and the effective monomer size from 2 Å to 4 Å. This points towards an affine deformation acting on all length scales similarly. This homogeneous in-plane deformation of the molecule is in agreement with the Gaussian-shaped out-of-plane monomer density profile extracted from the specular reflectivity analysis in sharp contrast to a box profile that would appear in case of squeezing of only the outer part (shell) of the molecule. We would like to stress here that the Gaussian-type nature of the average polymer conformation, in the here studied highly non-spherical state, is not intuitive as the sizes of $R_g^\perp$ undermatch the bulk $R_g$ of even the lowest molecular weight PSS reported in literature, to our knowledge, which is around 25 Å for a PSS chain of molecular weight $M_w = 14,000$ g/mol[20].

From specular NR a clear reduction in film thickness is observed for sprayed films compared to spin-coated and dipped ones. This change is accompanied by an increase of fractal scattering observed by SANS for the sprayed films. This fractal scattering bares similarities with the scattering from agglomerates in bulk PECs and could be explained by the increased amount of defects observed in NR by the smaller difference between the protonated and deuterated layer SLDs of the sprayed and spin-coated samples compared to the dipped ones. The SANS fractal scattering could also be observed in the OSS measurements in the form of Yoneda scattering in thick films as e.g. shown for a dipped sample in the Supplementary Information (see Supplementary Fig. 8), which could be fitted with correlation lengths on the order of several 1000 Å, slightly larger as compared to the SANS fits. All of this points towards a less relaxed conformation of the PSS chains in the sprayed films, which is likely due to quicker adsorption kinetics during preparation. It is known that PE adsorption takes place faster if the sample is agitated. Moreover, from quartz-crystal microbalance with dissipation (QCM-D) measurements it is known that PSS/PAH adsorption of one layer is finished after about 1 min in dipping conditions[11], while during spin-coating or spraying this process is much faster (a few seconds). On the other hand, QCM-D measurements on PSS/PDADMAC PEMS, a system which assembles closer to equilibrium with thicker PSS films, indicate that an hour is needed to finish the PSS adsorption process[35]. All of this points towards a memory of the microscopic structure of PEM films concerning the adsorption kinetics during preparation, resulting in more or less equilibrated structures.

The observation of off-specular reflection around the specular Bragg peaks indicates per se a partially phase separated polyanion/polycation mixing. This is opposed to monomer-scale intermixing as could be imagined in a scrambled egg-type complexation. The in-plane correlation length of this phase separation perfectly matches the in-plane molecular radius of gyration extracted from SANS, confirming that the intermixing between PSS and PAH happens on a molecular scale rather than on a monomer scale, pointing towards a ladder-type complexation process in LbL films and directly rejecting the suggestion that a fuzzy segment density profile means a scrambled egg-type complexation in PEMs[16].

Another important result from the OSS analysis is that the out-of-plane position of a PSS molecule compared to the mean interface between PSS and PAH is correlated over many layers, at least twenty layer pairs. This means that the electrostatic interaction is leading to a long-range order of molecules normal to the surface (in addition to the out-of-plane long-range order of the centers of mass of the PSS layers). This finding is in line with the fact that the extent of electrostatic interactions in PEMs, tested by ionization of a buried weak PE layer, was found to act beyond ten layers[36]. A similarity with bulk PEs is the electrostatic correlation peak related to long range order as evidenced in PE solutions[37] or complex coacervates[21].

In conclusion, by using advanced neutron scattering techniques we have determined the 3D nanoscopic structure of LbL assembled PEMs prepared by different methods, namely dipping, spray-assisted and spin-assisted assembly. We confirm, for the first time quantitatively and directly, that the conformation of a single PSS chain in a dipped multilayer film prepared from 2 M NaCl solutions is of a flattened coil form with an asymmetry factor of more than seven. This chain deformation seems to be of an affine-type acting on all length scales equally preserving a Gaussian nature of the segment density, which is particularly striking given the strong deformation of the chain. Moreover the volume occupied by the polymer chain in the PEM is equal to the volume occupied by a single chain in a bulk PEC. We also compared the out-of-plane radii of gyration for films prepared by spin-assisted assembly, dipping and spray-assisted assembly at two different salt concentrations and observed a clear trend towards less equilibrated structures with lower salt concentration. Similarly the preparation method influenced the structural asymmetry according to the sequence: spin-assisted assembly > spray-assisted assembly > dipping, testifying the memory of solid-like PEMs in regard to the deposition process. The less relaxed structure of sprayed samples compared to dipped ones was corroborated by higher fractal small angle neutron scattering, which we attribute to some kind of aggregation, also visible as increased amount of defects leading to a less pronounced stratification of polyanion and polycation layers. By using neutron reflectometry we also unambiguously show that the PSS monomer density profile is not completely flat revealing a Bragg peak for a sample with deuteration of every polyanion layer.

Finally, off-specular neutron reflectometry revealed that the complexation at the polyanion/polycation interface happens at the molecular level, rather than on the monomer level as could be expected from interdiffusion or scrambled-egg like complexation.

We hope that our study will contribute to a better understanding of PEM formation, internal structure and structure dependent materials properties. We show that the chosen method for PEM preparation has an influence on the polyelectrolyte conformation within the film. Establishing this fact is a prerequisite for further investigations and for better understanding the interplay between internal structure, chain mobility and materials properties. In this context it was assuring to find that in solid-like films like (PAH/PSS)$_n$ non-equilibrium structures, such as inserted deuteration-labeled layers, can be still observed after more than 15 years of storage at ambient conditions. Together with the fact that in some films polymer diffusion can be induced by weakening the electrostatic network in the film by exposure to salt solutions it is now becoming much clearer that there is a full toolbox for fine-tuning polyelectrolyte multilayers, which are essentially interfacial polyelectrolyte complexes, and their materials properties according to the needs of a specific application.

## Methods

### Materials

All chemicals were used as received unless stated otherwise. The chemical structures of the polyelectrolytes used for the build-up of the studied multilayer films are shown in Supplementary Fig. 1. The following polyelectrolytes were purchased from Sigma-Aldrich (Saint-Louis, USA): protonated poly(sodium 4-styrenesulfonate) sodium salt (PSS$_{h7}$, $M_w = 70,000$ g/mol) and poly(allylamine hydrochloride) (PAH, $M_w = 56,000$ g/mol and $M_w = 58,000$ g/mol). Deuterated poly(sodium 4-styrenesulfonate) sodium salt (PSS$_{d7}$, $M_w = 80,800$ g/mol - PdI $\leq 1.20$ and $M_w = 78,300$ g/mol - PdI $\leq 1.20$) were purchased from Polymer

Standard Service (Mainz, Germany). Branched poly(ethylenimine) (PEI, Lupasol HF, $M_w = 21,000$ g/mol) was purchased from BASF (Ludwigshafen am Rhein, Germany). $PSS_{h7}$ and $PSS_{d7}$ are polyanions and PEI and PAH are polycations. Sodium chloride (NaCl, ≥99.9% pure) was purchased from Carl Roth GmbH (Karlsruhe, Germany). Polyelectrolyte and rinsing solutions were prepared with ultrapure water, Milli-Q water (Milli-Q system, Millipore), with a resistivity of 18.2 MΩcm.

Substrates used for the deposition of the multilayer films for reflectometry and GISANS were one-side polished silicon wafers with an orientation (100) and a thickness of 700–775 μm. They were purchased from Wafernet Inc. (San Jose, USA). The substrates for transmission SANS were double-side polished float-zone grown intrinsic silicon wafers with a thickness of 500-550 μm purchased from Sil'tronix (Archamps, France) in order to suppress scattering from any impurities in the silicon[38]. Before the deposition of the films, all silicon wafers were rinsed with ethanol and Milli-Q water, followed by a drying step under compressed air. Then, they were treated with plasma in a plasma cleaner (PDC-002, Harrick Plasma, Ithaca, USA) for 3 min, to activate the surface. The wafers were used immediately after activation.

### Polyelectrolyte multilayer film build-up

All LbL films were deposited on activated silicon wafers with a first PEI layer adsorbed on the surface by dipping the substrate into a 1 mg/mL ($3 \times 10^{-3}$ monomol/L) PEI solution for 15 min, followed by three rinsing steps of 2 min into pure Milli-Q water, and dried under compressed air. Monomol corresponds to moles of the monomer repeat unit. LbL films were prepared from solutions at 0.6 mg/mL ($3 \times 10^{-3}$ monomol/L) for $PSS_{h7}$ or $PSS_{d7}$ and 0.27 mg/mL ($3 \times 10^{-3}$ monomol/L) for PAH dissolved in Milli-Q water containing 0.5 M or 2 M of NaCl.

Multilayer films were deposited on the substrates by dipping, spray- or spin-assisted LbL assembly, as described below. Samples for specular and off-specular reflectometry were built up by alternating 3, 4, 5 or 10 ($PSS_{h7}$-PAH) layer pairs with one ($PSS_{d7}$-PAH) layer pair, repeated up to 8 times and then capped with 0, 3 or 5 ($PSS_{h7}$-PAH) layer pairs. Additionally some samples for NR and SANS were composed of 70-100 layer pairs using a mixture of 30% $PSS_{h7}$ and 70% $PSS_{d7}$ and a PAH layer. Finally, one NR and GISANS sample consisted of 53 layer pairs composed of a mixture of 50% $PSS_{h7}$ and 50% $PSS_{d7}$ and a PAH layer $(PSS_{50h7-50d7}-PAH)_{53}$.

**Deposition by dipping.** Dipped films were prepared using a dipping robot by immersion of the substrates in the polyelectrolyte solutions for 12 min, with a lifting up of the samples out of the solutions every 2 min immediately followed by an immersion into the same solutions. This was done to induce a slight mixing of the solutions in order to improve the adsorption of the polyelectrolytes on the surface. The dipping in the polyelectrolyte solutions was followed by three rinsing steps of 2 min into pure Milli-Q water. After 1 min of dipping, a lifting up and down was also done for each rinsing step, to have a better rinsing. The samples were dried under compressed air every layer pair (after the polycation deposition) with an automated drying set-up mounted on the robot.

**Deposition by spray-assisted assembly.** LbL films prepared by spray-assisted assembly were built-up by three different methods: the manual spray with Air-boy cans, the automated spray with Aztek airbrushes and the automated spray with a grazing incidence angle using stainless steel nozzles. Air-boy cans were purchased from Carl Roth GmbH (Karlsruhe, Germany) and consist of containers with a manual pump to induce a pressure inside the solution chamber, allowing the spraying of the solutions. The nozzle has an internal diameter of 0.4 mm. Four different cans were used : one for the protonated polyanions, one for the deuterated polyanions, one for the polycations and one for the

water used to rinse the samples. The film deposition was performed at a distance of roughly 10 cm. Solutions were sprayed for 5 s, followed by a waiting time of 15 s. Then, the surfaces were rinsed with pure Milli-Q water sprayed for 5 s, also followed by a waiting time of 15 s. The samples were dried under compressed air either every layer for ellipsometry measurements (thickness evolution studies) or at the end of the build-up of the films (neutron scattering measurements). As the pressure inside the cans was decreasing during spraying, the cans were pumped every 2–3 spraying steps to keep a rather similar pressure during the whole build-up.

Aztek airbrushes (model A4809) were purchased from Kit Discount (Roquebrune-sur-Argens, France) and are composed of a solution inlet, a gas inlet and different nozzles with different internal diameters. The solutions were injected with pumps, and the gas and solution flux were computer-controlled. Three different pumps and airbrushes were used for the polyanions, the polycations and the rinsing water. The solutions were sprayed at a flux of 15 mL/min with a gas flux of 10 L/min or 20 L/min perpendicularly on a vertical substrate. The gas used was compressed air and the spray was done at a distance of roughly 10 cm. The nozzle had a internal diameter of 0.3 mm. The spraying was done in the same conditions than the spray with Airboy cans : 5 s of spraying of the solutions followed by a waiting time of 15 s, then a rinsing step of 5 s with 15 s of waiting. The samples were dried under compressed air either every layer for ellipsometry measurements (thickness evolution studies), or at the end of the film build-up (neutron scattering measurements).

For the grazing-incidence spraying stainless steel nozzles (model 1/4J-316S+SU26-316SS) were purchased from Spraying Systems Co. (Wheaton, USA) and consist of a solution inlet, a gas inlet and a nozzle with a fixed internal diameter. The solutions were injected with pumps, and the gas and solution flux were computer-controlled, with the same device as for the Aztek airbrushes. Three different pumps and airbrushes were used for the polyanions, the polycations and the rinsing water. Here the solutions were sprayed at a flux of 5 mL/min, with a gas compressed air flux of 30 L/min on a vertical substrate. The polyanion solution was sprayed at a distance of 2 cm with an angle of 15° between the surface of the samples and the spraying direction. The polycation solutions and the Milli-Q water for the rinsing were sprayed at a distance of roughly 10 cm perpendicularly to the surface. The spraying was done as followed : 20 s of spraying of the solutions followed by a waiting time of 15 s, then a rinsing step of 20 s with 15 s of waiting. The samples were dried under compressed air either every layer for ellipsometry measurements (thickness evolution studies) or at the end of the build-up of the films (neutron scattering measurements).

**Deposition by spin-assisted LbL assembly.** The spin-assisted LbL assembly was performed on a spin-coater WS-650-8B from Laurell Technologie Corporation (North Wales, USA). The spin-assisted LbL assembly was carried out at a rotation speed of 4000 rpm (rotation per minute) or 8000 rpm. The films were built-up by depositing ten drops of the solutions with a pipette on the rotating substrates, immediately followed by a rinsing step with 1 mL of pure Milli-Q water. Before the deposition of a new layer, the samples were left rotating until the removal of the solvent.

### Methods

Neutron scattering experiments were performed at several neutron institutions : the Institut Laue-Langevin (ILL), Grenoble, France, The High Flux Isotope Reactor (HFIR), Oak Ridge National Lab, TN, USA, the Forschungs-Neutronenquelle Heinz Maier-Leibnitz (FRMII), Garching, Germany, and the Laboratoire Léon Brillouin (LLB), Saclay, France, which are neutron reactor sources.

**Specular neutron reflectometry.** The instruments used for specular neutron reflectometry and GISANS are described below and the setups are summarized in Table 2.

**Table 2 | Setups used for the neutron reflectometry and GISANS measurements. Angular, wavelength and detector resolutions are given in FWHM**

| Setup reflectometry | Angle [°] | Collimation slits [mm] | Wavelength [Å] | Angular resolution | Wavelength resolution |
|---|---|---|---|---|---|
| D17-1 | 0.7 | 0.4/0.2 | 2–27 | $d\theta/\theta = 0.78\%$ | Variable |
|  | 3.8 | 1.6/0.8 |  |  |  |
| D17-2 | 0.7 | 0.4/0.2 | 2–27 | $d\theta/\theta = 0.78\%$ | Variable |
|  | 2.8 | 1.6/0.8 |  |  |  |
| FIGARO-1 | 0.624 | 0.216/0.216 | 2–20 | $d\theta/\theta = 0.9\%$ | $d\lambda/\lambda = 0.82\%$ |
|  | 2 | 0.7/0.216 |  |  |  |
| FIGARO-2 | 0.622 | 0.216/0.216 | 2–20 | $d\theta/\theta = 0.9\%$ | $d\lambda/\lambda = 0.82\%$ |
|  | 2 | 0.7/0.216 |  |  |  |
| N-REX+ | 0.1–2.5 | 1/1 | 4.3 | $d\theta = 0.029°$ | $d\lambda/\lambda = 3\%$ |
| SuperADAM | 0–5 | 3/3 | 4.4 | $d\theta = 0.01°$ | $d\lambda/\lambda = 0.6\%$ |

| Setup GISANS | Angle [°] | Collimation slits [mm] | Wavelength [Å] | Sample-to-detector distance [mm] | Angular resolution | Wavelength resolution | Detector resolution |
|---|---|---|---|---|---|---|---|
| FIGARO-3 | 1 | 6.667 / 3.333 | 2–20 | 2831 | $d\theta_y = 0.118°$ | $d\lambda/\lambda = 7\%$ | $2.2 \times 4.7$ mm$^2$ |
|  |  |  |  |  | $d\theta_z = 0.134°$ |  |  |

The D17[24] reflectometer at ILL with a vertical sample surface geometry was used in time-of-flight (ToF) mode using a wavelength range from 2.5 Å to 27 Å with the setups named D17-1 and D17-2 in Table 2. The 2D multidetector allowed the simultaneous measurement of specular and off-specular scattering. Specular reflectivities were reduced using COSMOS[39] and off-specular maps were reduced using Överlåteren[40].

On the ToF reflectometer FIGARO[41] at ILL with horizontal sample geometry a wavelength band between 2 and 20 Å was used. The two dimensional multitube detector allows the measurement of specular and off-specular reflectivity, as well as GISANS measurements. The detector has a size of 25 x 48 cm² and a resolution of 2.2 x 4.8 mm² (FWHM) at a distance of 2.8 m from the sample. We used FIGARO to perform neutron reflectometry (specular and off-specular) and GISANS measurements, with the setups FIGARO-1, FIGARO-2 and FIGARO-3. The specular NR was reduced using COSMOS[39] and the OSS maps with self-written IDL code.

N-REX+[42] at FRM2 is an angle dispersive fixed wavelength (4.4 Å) reflectometer with horizontal sample geometry. A 20 x 20 cm² position sensitive detector was used allowing for specular and off-specular reflectivity measurements. We carried out neutron reflectometry (NR and OSS) on N-REX+, with the setup N-REX+. Data were reduced with home-built programs.

SuperADAM[43] at ILL is an angle dispersive reflectometer with a vertical sample surface geometry and a two dimensional position sensitive detector allowing for specular and off-specular reflectivity measurements. We performed neutron reflectometry (NR and OSS) on SuperADAM, with the setup SuperADAM. Data were reduced by home-written code in IgorPro.

Specular NR data were analysed using Motofit[44]. For the fitting of the reflectivity, we used a box model to describe our films, in which each box (slab or layer) was described by three structural parameters : the thickness, the SLD and the roughness. In all polyelectrolyte films measured by neutron reflectometry, two sub-layers were present on the silicon crystal : a silicon oxide layer (SiO$_2$ layer) and a PEI layer. The thickness of both layers was determined by ellipsometry and the thicknesses of the SiO$_2$ layer were adjusted to have a better fit, but with a maximum variation of 2-3 Å from the value determined by ellipsometry. The SLD and the roughness were fixed at 3.15 x 10$^{-6}$ Å$^{-2}$ and 4 Å respectively, which are the common values used for the SiO$_2$ layer for this type of samples[27]. The SLD of the PEI layer was assumed to be identical to the SLD of the layer deposited on it, and was fitted with it. As there is no difference between the SLDs of the PEI layer and the layer above, the roughness of the PEI layer, which corresponds to the

interface between the two layers, has no influence on the fit and cannot be fitted. So it was fixed at 9 Å, as done previously[27]. In the fitting procedure the starting values for the SLDs and roughnesses were determined from the average values found in the previous neutron reflectometry measurements on (PSS-PAH)$_n$ films and the starting thicknesses were determined from measurements of the film thicknesses by ellipsometry. Then, the parameters were fitted first manually, followed by a numerical fit using the Levenberg-Marquardt algorithm. Errors for non-fixed parameters were calculated using the create local chi2map for requested parameter option of Motofit. 2D Chi$^2$ maps were created for two correlated parameters (each non-fixed parameter was correlated to at least one other parameter) and the errors for each parameter were determined by a 5% increase in $\chi^2$. We performed fits of the specular reflectivity curves by considering an inhomogeneous structure, by separating the films in three parts, the bottom layers next to the silicon substrate, the top layers next to the surface and the bulk layers in the middle of the films. The bottom layers part of the films is composed of the three first (PSS-PAH) layer pairs, deposited on the PEI layer. We assumed that these layers have a smaller thickness per layer pair than the bulk layers and a smaller roughness at the interfaces. The top layers part is composed of the last (PSS-PAH) layer pair. The bulk layer part is composed of the rest of the film. The structure is considered homogeneous in this bulk layers part, which means that the thickness per layer pair, the SLD and the roughness are the same for all the layers.

We found that the roughness and thickness of the layers were not independent parameters, but rather linked to each other describing a Gaussian distribution, rather than layers with sharp interfaces. This not only corroborates that the polyelectrolyte layers are monolayers, but also opens up the possibility to extract the single chain extension perpendicular to the interface. The probability density of an ideal polymer coil corresponds to a Gaussian-type distribution[34]:

$$P(z) \sim \exp\left(\frac{-z^2}{4(R_g^{\perp})^2}\right) \tag{1}$$

with the radius of gyration in the perpendicular direction $R_g^{\perp}$. Following along this line we fitted a Gaussian function to the SLD profile of a single PSS layer for every sample as shown in Fig. 1 and extracted the radius of gyration. The results are tabulated in Table 1.

**Small Angle Neutron Scattering (SANS).** The SANS measurements were performed by transmission through the silicon supported thin films on three different machines: On D11 at ILL[45] using two detector

distances (8 m and 34 m) at 6 Å wavelength, on D22 at ILL using only one detector distance of 17.6 m at 6 Å and on GP-SANS at HFIR[46] using 6 Å wavelength at 19.3 m sample-to-detector distance. The data at ILL were reduced using LAMP[47] and at HFIR using home-built macros written in IgorPro (Wavemetrics Inc.). All data was normalized to the total thickness determined by ellipsometry yielding absolute macroscopic cross-sections. Due to the very thin films on the order of several 100 nm the counting times for a whole SANS curve varied between 3 and 6 h.

The theoretical scattering cross section $\frac{d\sigma}{d\Omega}$ of a polymer blend of deuterated and protonated chains is dominated by the polymer form factor $F(\mathbf{q})$ in the measured q-range. Under the assumption that the polymerization $n_P$, structure and monomer volume $v_P$ is the same for the deuterated and protonated chains, and there is perfect mixing of the two species, the scattering cross section is:

$$\frac{d\sigma}{d\Omega} = (N_{bD} - N_{bH})^2 \Phi_D \Phi_H n_P v_P F(\mathbf{q}), \tag{2}$$

with the volume fraction of the deuterated and protonated chains, $\Phi_D$ and $\Phi_H$, respectively, the scattering length densities $N_{bD}, N_{bH}$, respectively, for the deuterated and protonated chains. For the parameters in this study, $N_{bD} = 3.9*10^{10}$ cm$^{-2}$, $N_{bH} = 1.55*10^{10}$ cm$^{-2}$, if using the density of 1 g/cm$^{-3}$ of PEMs[48] and no water condensation, $\Phi_D = 0.3$, $\Phi_H = 0.7, v_P = 4.3*10^{-22}$ cm$^3$ and $n_P = 340$, the pre-factor in front of the form factor is 17 cm$^{-1}$. From specular fits the SLD of the PEs was estimated to be lower due to hydration, down to $N_{bD} = 2.7*10^{10}$ cm$^{-2}$ and $N_{bH} = 1.2*10^{10}$ cm$^{-2}$, taking this file we get a scaling of 6.9 cm$^{-1}$.

The quantitative SANS analysis was performed either by using the fitting program SASFit[49] version 0.94.11 or by using a hand-written code as explained later. In both cases the total scattering function was composed of the sum of several contributions. In SASFit the low-q contribution was fitted as a mass fractal with an exponential cut-off, having the size of the individual scatterers, the size of the aggregate and the fractal dimension as free fitting parameters. The mid q-range was assumed either as a sum of a Debye function with the radius of gyration and the scattering contrast as fitting parameters and a freely jointed chain of rods with the Kuhn length and scaling as fitting parameters, or, a Kholodenko wormlike chain with the scattering contrast, the Kuhn length and the radius of gyration as fitting parameters. In any case for the highest q-range a constant background was added as well.

Alternatively a hand-written formula was also used to fit the SANS curves following the approach presented in ref. 32, with an additional contribution for the fractal scattering (same as in SASFit):

$$1 + (D*\Gamma(D-1)*\sin((D-1)*\tan^{-1}(q*\xi)))/((q*r_0)^D*(1+1/(q*\xi)^2)((D-1)/2)), \tag{3}$$

with $\xi$ being the size of the agglomerate, $D$ the fractal dimension and $r_0$ the size of the individual scatterer.

For the low q scattering ($q*l_p < 2$) the fit function was:

$$I_0/N*(R_g^2*3/(b*l_p)*(2/x^2*(x-1+\exp(-x))+2*l_p^2/(3*R_g^2)*(4/15+7/(15*x) \\ -(11/15+7/(15*x))*\exp(-x)))), \tag{4}$$

with $x = (R_g*q)^2$, $R_g$ the radius of gyration, $l_p$ the persistence length, $b$ the monomer size, $I_0$ the scattering contrast and $N$ the polymerization. For the mid q-range ($2 < q*l_p < 4$) the fit function was:

$$I_0/(N*q^2*b*l_p)*(6+0.5470*(q*l_p)^2 - .01569*(q*l_p)^3 - 0.002816*(q*l_p)^4), \tag{5}$$

and for the high q-range ($q*l_p > 4$):

$$I_0/N*\pi/(q*b) + 2/(3*q^2*b*l_p). \tag{6}$$

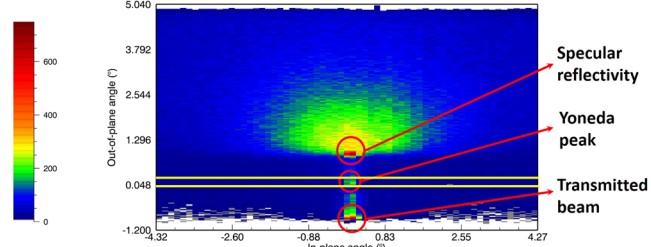

**Fig. 5 | Grazing Incidence Neutron Small Angle Scattering.** GISANS pattern (arbitrary counts on linear color scale) of the dipped film from 2 M NaCl solution with 53 layer pairs at a wavelength of 3.2 Å.

The function corresponding to the right q-range was multiplied with the fractal scattering function presented above and a constant background was added. The resulting function was then fitted to the SANS curves using the least squares algorithm in Igor Pro (Wavemetrics Inc.).

**Grazing Incidence Small Angle Neutron Scattering (GISANS).** For the GISANS measurements we followed the work of J. Kraus et al.[50], who studied the radius of gyration of polystyrene chains in films of polystyrene. The films were composed of layers with a mixture of 50% deuterated and 50% protonated PSS. The measurements were performed on FIGARO at ILL with the setup FIGARO-3 from Table 2. As each measurement was done in TOF mode, data were treated for every wavelength individually. Data were treated to remove the background reflection from a bare Si surface and normalized to the incident beam intensity. Then, a horizontal cut (in the y direction, parallel to the surface) at the height of the Yoneda peak, as shown in Fig. 5 by the yellow lines was extracted from the 2D image for each wavelength individually. For the final analysis we used data at five mean wavelengths : 4 Å (integration over 3.6–4.4 Å), 5 Å (integration over 4.5–5.5 Å), 7 Å (integration over 6.3–7.7 Å), 9 Å (integration over 8.1–9.9 Å) and 11 Å (integration over 10–12 Å). Due to optical effects of the grazing incident geometry the absolute scattering intensities are in general wavelength dependent[33]. Especially close to the critical momentum transfers (Yoneda peak) of the incoming or outgoing wave vector the reflected signal is highly enhanced. Therefore we analyzed the data at different wavelengths individually. It turned out that data recorded at 4–7 Å collapsed onto a master curve, but the longer wavelengths showed a clear increase in intensity due to the proximity of the critical momentum transfer of the incoming wave vector (~20 Å) at the chosen reflection angle (1°).

**Off-specular Neutron Scattering (OSS).** Off-specular data maps were qualitatively compared between different samples either using the raw detector images, or after normalizing to the incident beam spectrum and transferring into momentum transfer space[51].

Quantitative fits of the OSS patterns were performed using the algorithm described in ref. 33. Therefore the specular reflectivity model was used with two in-plane SLD correlations leading to off-specular scattering: the first one was correlated roughness at the deuterated polymer/ protonated polymer interfaces modeled with a Gaussian in-plane correlation length, which turned out to be in the tens of nm range. The equivalent radius of gyration was extracted from this correlation length by using eq. (1). This roughness was assumed to be also correlated between neighboring interfaces, with an exponentially decaying out-of-plane correlation length. This perpendicular correlation length turned out to be on the order of the total film thickness for all investigated samples leading to Bragg sheets as seen in e.g. Fig. 4a.

The second contribution to OSS was assumed to be in-plane inhomogeneities across the whole layer brought to light by the nominal SLDs from specular fits deviating form the pure deuterated and

protonated polymer SLDs. Here the pure polymer SLDs were assumed to be $3.7*10^{-6}Å^{-2}$ for the partially hydrated d-PSS and $0.3*10^{-6}Å^{-2}$ for the partially hydrated protonated polymers and the concentration of the defects was calculated according to the fitted nominal SLDs from the specular NR analysis. The form of these defects was modeled as an exponential in-plane correlation length similar to the SANS fractal scattering model. The result of this contribution was OSS scattering around the critical edge known as Yoneda peaks (see e.g. Supplementary Fig. 8). To fit the OSS data correlation lengths in the hundreds of nm range had to be assumed, slightly larger than the size of agglomerates extracted from the SANS analysis.

**Ellipsometry.** A spectroscopic ellipsometer (SENpro, SENTECH Instrument GmbH) was used to determine the refractive indices and the total thickness of the films. It was operated at a fixed angle of 70° and a wavelength range of 3700 Å - 10500 Å.

## Data availability
The ILL raw and processed data generated in this study have been deposited in the ILL database under the following links: https://doi.org/10.5291/ILL-DATA.9-11-1645 and https://doi.org/10.5291/ILL-DATA.9-11-1684. The data from the other neutron sources are available upon request from the corresponding author.

## Code availability
The code used to calculate the off-specular scattering maps is available after registration to the ILL user club on code.ill.fr/gutfreund/dwba-again.

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

## Acknowledgements

We thank the ILL for beam time on D11, D17, FIGARO and S-ADAM within experiments https://doi.org/10.5291/ILL-DATA.9-11-1645 and https://doi.org/10.5291/ILL-DATA.9-11-1684. We thank Ralf Schweins for his help during the SANS measurements on D11. We also acknowledge preliminary test time on D22 with the help of Lionel Porcar. We acknowledge beam time on GP-SANS at HFIR and thank Yuri Melnichenko for his help. We thank MLZ for beam time on N-Rex+ with the help of Yury Khaydukov. We also acknowledge test time on AMOR at PSI. The PhD work of C.H. was co-financed by ILL and CNRS through an IRTG grant (contract ILL-1142-CNRS). We thank Jean-Louis Barrat for cross-reading the manuscript and his helpful comments. We thank the PSCM for pre-characterization of samples.

## Author contributions

P.G. and C.H. performed most of the experimental work and data analysis. G.F. participated in NR experiments. C.H. and M.T. prepared the samples. P.G., O.F. and G.D. designed the study. P.G. wrote the first version of the manuscript. All authors reviewed and edited the final version of the manuscript.

## Competing interests

The authors declare no competing interests.
