## [Peer Review File · Nature Communications]

Molecular conformation of polyelectrolytes inside Layer-by-Layer assembled filmsReviewers' Comments:

Reviewer #1:

Remarks to the Author:

Nature Communications Gutfreund et al

This paper reports investigation of the molecular conformation of polyelectrolytes in multilayer thin films (PEMs) using the power of various neutron scattering techniques enabled by deuterium labelling of one of the PE components. The neutron expertise is provided by scientists based at the Institut Laue-Langevin in Grenoble and the chemical expertise for sample preparation is based at the Institut Charles Sadron in Strasburg. Neutron reflectivity experiments from samples in which some whole layers of one of the two PE components are fully deuterated yield detailed information on the layer structure, and the interfaces between PE components, while samples in which this PE component contains a mix of deuterated and normal PE molecules when observed by small angle scattering yield the shape and size of individual molecules. The result is a genuine "tour de force" demonstrating the power of SANS and neutron reflectivity measurements to elucidate molecular detail in these complex PEM systems. The paper should certainly be published

The authors investigated three different preparation routes for the multilayered structures using poly(styrene -sulphonate) sodium salt (PSS) and its perdeuterated analogue as the labelled PE (anion) and poly (allylamine hydrochloride) (PAH) as the second PE component (cation). The three layer by layer (LbL) preparation routes were dipping, spraying, and spin coating. For the first time the nature of the PSS conformation was revealed to be a highly asymmetric flattened coil with the in-layer diameter 7 or more times larger than the dimension perpendicular to the layers. This flattened coil shape relates to the symmetric shape in a solution before dipping which has suffered affine deformation as the molecules get flattened into the layers. The asymmetry is most pronounced in the dipped samples where the conformation has more time to equilibrate than in spun or sprayed samples. As well as the asymmetry of the molecules the results show complexation of anions and cations occurs at a molecular rather than monomer level and demonstrate the existence of a long range order of these molecules normal to the substrate surface.

Comments for consideration by the authors

1 It is reasonable that the paper does not include details of the extensive literature on preparation and applications of multilayer PE films, however in their Conclusions section, the authors note "We hope this study will lead to a better understanding of PEM behaviour and reliability". I think clarification of the ways in which the reported molecular structure would be expected to affect PEM behaviour and reliability would be useful for the reader to contextualise the results. Could the authors include a short section?

2 The reporting of the data makes the point that results have been obtained on several instruments and a number of neutron sources. This has clearly happened over a number of years and during that time there has been considerable developments in the neutron techniques and data analysis. However I note that the section on neutron techniques does not include a mention of The LLB (Laboratoire Leon-Brillouin) while Figures S1 and S2 appear to have been obtained in 2007 from that laboratory. Are these old data which have already been published? Perhaps the authors could put these data in context.

Reviewer #2:

Remarks to the Author:

This paper applies a wide range of advanced neutron scattering techniques to study conformations of adsorbed polyelectrolytes in a well-known PSS/PAH polyelectrolyte multilayer systems. Despite the old system, new and fundamentally important observations are reported. The first one is the quantitative measurements of the asymmetry of the assembled polymer chains. The second one is the demonstration of the potential of GI and transmission SANS to determine lateral chain dimensions. Finally, the fractal analysis of transmission SAXS data has revealed, to my knowledge for the first time, significant changes in internal structure of spray-, spin and dip-deposited multilayers. I believe

that the collection of these finding has a potential to have a significant impact in the community of polymer scientists and engineers.

As the first critical comment, the authors should make it clear, including in the abstract and conclusions, that the determined value of the chain asymmetry refers only to one specific deposition condition (2M NaCl). It would be also helpful to comment how these types of measurements can be useful to other systems and other ionic strengths. For example, have the authors tested lower ionic strengths to observe a larger than 7 asymmetry values? In addition, the discussion of the random walk should be revised, as conformation of the adsorbed chains cannot be strictly following the random walk statistics, as the adsorption sites break the randomness of the statistical walk. What is observed is the Gaussian distribution in the z direction, but at the same time this might not necessarily mean that the entire polymer chain is performing a random walk.

Reviewer #3:

Remarks to the Author:

Review Layer-by layer

The authors present a thorough study on the molecular configuration of layered films of polyelectrolytes using different techniques using neutron scattering and reflectometry. They observe trends for the different techniques and different salt concentrations. The article is clearly written, with a good structure and a clear message. The amount of data from many different neutron sources and instruments is impressive. The data allows to come to the clear conclusion that the conformation is like a pancake. The effect of construction technique and of salt are well interpreted. Literature is well cited. After one minor addition I recommend this article for your journal.

The transmission SANS presented in Fig. 2 is measured on a film with ~ 80 layers with a thickness of 40 Ångström, which gives a total thickness of 0.32 μm , while a normal sample thickness in SANS is 1 mm. How is it technically possible to get a good signal with a sample that is more than a factor 1000 thinner than under normal circumstances? If the authors can explain this.

Then I have 2 minor remarks:

The Table 2 should be of higher quality. The contents are very relevant and should be easily readable. In the supplementary material on page S4 the first word of the section title "Preparaion" should be corrected.

Remarks of the reviewers are in blue and our responses to the remarks are in black color.

We would like to thank the three reviewers for taking the time for carefully reading our manuscript and for providing us with detailed and critical questions/suggestions that were very helpful for improving the manuscript.

Reviewer #1:

- 1) "It is reasonable that the paper does not include details of the extensive literature on preparation and applications of multilayer PE films, however in their Conclusions section, the authors note "We hope this study will lead to a better understanding of PEM behaviour and reliability". I think clarification of the ways in which the reported molecular structure would be expected to affect PEM behaviour and reliability would be useful for the reader to contextualise the results. Could the authors include a short section?"

We have extended the discussion of possible implications of the reported results on PEM behaviour by including the following section in the conclusions: "We hope that our study will contribute to a better understanding of PEM formation, internal structure and structure dependent materials properties. We show that the chosen method for PEM preparation has an influence on the polyelectrolyte conformation within the film. Establishing this fact is a prerequisite for further investigations and for better understanding the interplay between internal structure, chain mobility and materials properties. In this context it was assuring to find that in solid-like films like (PAH/PSS)_n non-equilibrium structures, such as inserted deuteration-labelled layers, can be still observed after more than 20 years of storage at ambient conditions. Together with the fact that in some films polymer diffusion can be induced by weakening the electrostatic network in the film by exposure to salt solutions it is now becoming much clearer that there is a full toolbox for fine-tuning polyelectrolyte multilayers, which are essentially interfacial polyelectrolyte complexes, and their materials properties according to the needs of a specific application."

- 2) "The reporting of the data makes the point that results have been obtained on several instruments and a number of neutron sources. This has clearly happened over a number of years and during that time there has been considerable developments in the neutron techniques and data analysis. However I note that the section on neutron techniques does not include a mention of The LLB (Laboratoire Leon-Brillouin) while Figures S1 and S2 appear to have been obtained in 2007 from that laboratory. Are these old data which have already been published? Perhaps the authors could put these data in context."

These data, presented in the Supplementary Information, were indeed obtained at the LLB and partially published already as indicated in the accompanying text. We have added this information to the Methods part to make it clearer.

Reviewer #2:

- 1) "As the first critical comment, the authors should make it clear, including in the abstract and conclusions, that the determined value of the chain asymmetry refers only to one specific deposition condition (2M NaCl). It would be also helpful to comment how these

types of measurements can be useful to other systems and other ionic strengths. For example, have the authors tested lower ionic strengths to observe a larger than 7 asymmetry values?"

We have made it clear now in the abstract and conclusions that the in-plane chain size was only determined for the 2M NaCl condition. For the other ionic strengths only out-of-plane data is reliably accessible as the in-plane scattering is too weak as stated in the SANS Results section: "For these investigations we will use films produced at relatively high salt concentration (2M NaCl) in order to increase the film thickness as the scattering intensity scales with the square of the film thickness and is generally very low in these types of measurements."

- 2) "In addition, the discussion of the random walk should be revised, as conformation of the adsorbed chains cannot be strictly following the random walk statistics, as the adsorption sites break the randomness of the statistical walk. What is observed is the Gaussian distribution in the z direction, but at the same time this might not necessarily mean that the entire polymer chain is performing a random walk."

We thank the referee for pointing this out and we fully agree. We have changed the phrasing of random-walk into Gaussian-type density distribution throughout the manuscript.

Reviewer #3:

- 1) "The transmission SANS presented in Fig. 2 is measured on a film with ~80 layers with a thickness of 40 Ångström, which gives a total thickness of 0.32 μm, while a normal sample thickness in SANS is 1 mm. How is it technically possible to get a good signal with a sample that is more than a factor 1000 thinner than under normal circumstances? If the authors can explain this."

We have added a sentence in the Methods section to explain how very thin films can be measured on high flux neutron sources like ILL and HFIR: "Due to the very thin films on the order of several 100 nm the counting times for a whole SANS curve varied between 3 and 6 h." Another important detail for success of these kind of measurements is stated in the Methods part about sample preparation: "The substrates for transmission SANS were double-side polished float-zone grown intrinsic silicon wafers with a thickness of 500-550 micrometer purchased from Sil'tronix (Archamps, France) in order to suppress scattering from any impurities in the silicon [38]."

- 2) "The Table 2 should be of higher quality. The contents are very relevant and should be easily readable."

We have remade this table to make it better readable.

- 3) "In the supplementary material on page S4 the first word of the section title "Preparaion" should be corrected"

We have corrected this typo.

Reviewers' Comments:

Reviewer #1:

Remarks to the Author:

I have read the response of the authors to all three referees and seen the relevant modifications in their revised text. I am content that my own comments have been satisfactorily dealt with and i believe so also have those of reviewers 2 and 3. I believe the paper is now ready for publication.

Reviewer #2:

Remarks to the Author:

The authors have provided a satisfactory response to my comments, and I support publication of this work as is.

Reviewer #3:

Remarks to the Author:

My concerns with the previous manuscript are resolved. I recommend the article for publication.

We thank the reviewers for taking the time for carefully reading the revised version of our manuscript. We are very grateful that all three referees recommend the publication now:

Reviewer 1) "I am content that my own comments have been satisfactorily dealt with and i believe so also have those of reviewers 2 and 3. I believe the paper is now ready for publication."

Reviewer 2) "The authors have provided a satisfactory response to my comments, and I support publication of this work as is."

Reviewer 3) "I recommend the article for publication."